# Lessons from the meiotic recombination landscape of the ZMM deficient budding yeast *Lachancea waltii*

Fabien Dutreux[1], Abhishek Dutta[1], Emilien Peltier[1], Sabrina Bibi-Triki[1], Anne Friedrich[1], Bertrand Llorente[2]*, Joseph Schacherer[1,3]*

**1** Université de Strasbourg, CNRS, GMGM UMR 7156, Strasbourg, France, **2** CNRS UMR7258, INSERM U1068, Aix Marseille Université UM105, Institut Paoli-Calmettes, CRCM, Marseille, France, **3** Institut Universitaire de France (IUF), Paris, France

☉ These authors contributed equally to this work.
* bertrand.llorente@inserm.fr (BL); schacherer@unistra.fr (JS)

**Data Availability Statement:** Sequence data deposited in the European Nucleotide Archive under the accession number PRJEB39767.

## Abstract

Meiotic recombination is a driving force for genome evolution, deeply characterized in a few model species, notably in the budding yeast *Saccharomyces cerevisiae*. Interestingly, Zip2, Zip3, Zip4, Spo16, Msh4, and Msh5, members of the so-called ZMM pathway that implements the interfering meiotic crossover pathway in *S. cerevisiae*, have been lost in *Lachancea* yeast species after the divergence of *Lachancea kluyveri* from the rest of the clade. In this context, after investigating meiosis in *L. kluyveri*, we determined the meiotic recombination landscape of *Lachancea waltii*. Attempts to generate diploid strains with fully hybrid genomes invariably resulted in strains with frequent whole-chromosome aneuploidy and multiple extended regions of loss of heterozygosity (LOH), which mechanistic origin is so far unclear. Despite the lack of multiple ZMM pro-crossover factors in *L. waltii*, numbers of crossovers and noncrossovers per meiosis were higher than in *L. kluyveri* but lower than in *S. cerevisiae*, for comparable genome sizes. Similar to *L. kluyveri* but opposite to *S. cerevisiae*, *L. waltii* exhibits an elevated frequency of zero-crossover bivalents. Lengths of gene conversion tracts for both crossovers and non-crossovers in *L. waltii* were comparable to those observed in *S. cerevisiae* and shorter than in *L. kluyveri* despite the lack of Mlh2, a factor limiting conversion tract size in *S. cerevisiae*. *L. waltii* recombination hotspots were not shared with either *S. cerevisiae* or *L. kluyveri*, showing that meiotic recombination hotspots can evolve at a rather limited evolutionary scale within budding yeasts. Finally, *L. waltii* crossover interference was reduced relative to *S. cerevisiae*, with interference being detected only in the 25 kb distance range. Detection of positive inference only at short distance scales in the absence of multiple ZMM factors required for interference-sensitive crossovers in other systems likely reflects interference between early recombination precursors such as DSBs.

**Funding:** This work was supported by the Agence Nationale de la Recherche (ANR-18-CE12-0013) to BL and JS as well by the European Research Council (ERC Consolidator Grant 772505) to JS. The funders had no role in study design, data collection and analysis, decision to publish, or preparation of the manuscript.

**Competing interests:** The authors have declared that no competing interests exist.

## Author summary

Studying non-model species is relevant to understand better biological processes by shedding light on their evolutionary variations. Here we chose the non-model budding yeast *Lachancea waltii* to study meiotic recombination. In sexually reproducing organisms, meiotic recombination is essential for gamete formation, and it shuffles parental genetic combinations notably by crossovers that cluster in hotspots at the population level. We found remarkable variations compared to both the canonical *Saccharomyces cerevisiae* model, also known as the baker's yeast, and another close relative *Lachancea kluyveri*. *L. waltii* meiotic chromosomes are frequently devoid of crossover, suggesting the existence of an alternative mechanism that efficiently ensures gamete formation. In addition, in line with the *L. waltii* specific loss of several genes controlling interference between meiotic crossovers, a process promoting even crossovers spacing, we found only residual crossover interference in *L. waltii*. This residual crossover interference is likely the result of the modest interference existing between recombination precursors that is often disregarded. Finally, while crossover hotspots were found to be remarkably stable across the Saccharomyces species, we found here that they are not conserved between the Lachancea and the Saccharomyces clades. This shows that crossover hotspots can evolve at a rather limited evolutionary scale within budding yeasts.

## Introduction

Homologous recombination is a ubiquitous DNA repair process that generates crossover and noncrossover recombinants [1]. In most species, crossovers resulting from meiotic recombination are essential for accurate homologous chromosome segregation at the first meiotic division and hence fertility [2]. In addition, by promoting shuffling of parental alleles, both crossover and noncrossover recombinants promote diversity [3]. Meiotic recombination is, therefore, under strong selective constraint.

Meiotic recombination results from repairing programmed DNA double strand breaks (DSBs) made at leptotene by a multi-protein complex. The catalytic subunit is the topoisomerase II-like trans-esterase Spo11 protein [4,5]. DSB formation is highly regulated. In many organisms, it eventually results in DSB hotspots in nucleosome-depleted regions within loops of chromatin generated by the compaction of the meiotic chromosomes. DSB hotspots and corresponding recombination hotspots are frequently enriched in gene promoters and other functional elements in non-metazoans organisms that all lack the PRDM9 methyl transferase [4,6–11]. When PRDM9 is present, DSB and recombination hotspots are determined by its sequence specific DNA binding provided by a rapidly evolving minisatellite encoding an array of zinc fingers [12–15]. Hence, PRDM9 determined DSB and recombination hotspots are particularly fast evolving, unlike those determined by functional elements that are more stable across evolution [4,10,16–21]. These two drastically different evolutionary dynamics of DSB and recombination hotspots are the only known so far.

In addition to the determining effect of DSBs, recombination events distribution along chromosomes undergoes other selective constraints, notably because some configurations are at risk. Due to its repetitive nature, DSB formation and recombination are at risk in the rDNA locus [22]. Crossover formation near centromeres may threaten chromosome segregation during meiotic divisions I and II and are therefore repressed, primarily by preventing DSB formation nearby [23]. In different yeasts and fungi, recombination is prevented near the mating-type locus, sometimes over several hundreds of kilobases corresponding to an entire

chromosome arm [24]. This may be linked to the risk of rendering the mating type locus homozygous and hence perturbing allele frequency of the two mating types within the population. Last but not least, while a single crossover combined with sister chromatid cohesion allows proper tension to be generated between homologous chromosomes at meiosis I, the presence of a second crossover at the vicinity of the first one would likely prevent such tension from being generated due to restrained cohesion between sister chromatids, hence promoting chromosome missegregation at meiosis I. Crossovers, therefore, tend not to form at the vicinity of one another as a result of a mechanism called crossover interference [25]. The molecular mechanism of crossover interference is so far unknown. In *S. cerevisiae*, it involves topoisomerase II and is independent of the synaptonemal complex [26]. However, SC central region proteins are essential for imposing crossover interference in *Caenorhabditis elegans* and in *Arabidopsis thaliana*, suggesting the existence of different layers of crossover control which contributions vary between species [27–32]. The strength of crossover interference also varies among organisms. Only one crossover per chromosome is formed in *C. elegans* due to a crossover interference strength that exceeds the size of the chromosomes [30–33]. On the contrary, crossover interference hardly spreads above 100 kb in *Saccharomyces cerevisiae* [33]. In *S. cerevisiae*, interfering crossovers, also named type I crossovers, are generated through the so-called ZMM pathway, which historically comprises Zip1, Zip2, Zip3, Zip4, Spo16, Msh4, Msh5 and Mer3 [34]. The Pph3 phosphatase subunit and the proteasome subunit Pre9 are more recently identified members of the ZMM pathway [35]. The ZMM pathway relies on the nuclease activity of the Mlh1-Mlh3 heterodimer for resolution of recombination intermediates exclusively into crossovers [36–40]. There are also non-interfering crossovers or type II crossovers in *S. cerevisiae* [41]. They are less abundant than type I crossovers and are formed independently of the ZMM proteins through the resolution of recombination intermediates by the structure specific nucleases Mus81, Yen1 and Slx1 [42,43]. Remarkably, while the ZMM proteins are conserved from *S. cerevisiae* to mammals, they have been lost several times independently during evolution. They are absent from *S. pombe* and from different species of the *Saccharomycotina* subphylum. They have been lost in *Eremothecium gossypii* after its divergence from *Eremothecium cymbalariae* [44]. They have been partially or completely lost in the *Candida* clade [45]. And finally, all the ZMM genes except *ZIP1* and *MER3* have been lost in the *Lachancea* clade with the exception of *Lachancea kluyveri* species [46]. Either non-orthologous gene displacement systematically allowed to fulfil the ZMM crossover formation and patterning functions in species that lost all or some of the ZMM members, or alternative regulation of crossover formation and patterning exist in such species.

The genomic era now allows precise determination of genome-wide meiotic recombination landscapes and frequencies virtually in any species. This information is important to better understand the mechanisms controlling crossover formation, crossover patterning, and to understand how recombination landscapes and frequencies evolve, a matter poorly explored despite frequent variations between individuals, populations, sexes and taxa [47,48]. Genome-wide meiotic recombination landscapes have been determined in many organisms including yeasts species, flies, plants and mammals [49–59]. Besides a few exceptions for *Drosophila* such information is rarely available for numerous individuals from a same species or from closely related species precluding answering precisely evolutionary questions [50,60]. In this context, we recently explored the recombination landscape of *Lachancea kluyveri*, a protoploid yeast species that diverged from the *Saccharomyces* genus more than 100 million years ago and we found striking differences with *S. cerevisiae* [61]. These variations include a lower recombination rate (1.6 *vs* 6.0 crossover/Mb), a higher frequency of chromosomes segregating without any crossover and the absence of recombination on the entire chromosome arm containing the sex locus [61].

In the continuity of this work, here we determined the genome-wide recombination landscape of another *Lachancea* species, *Lachancea waltii*, that lacks *MLH2* and most ZMM genes except orthologs of *ZIP1* and *MER3* (S1 Table) [46]. With 3.4 crossovers/Mb, we found that the *L. waltii* recombination rate is intermediate between the *S. cerevisiae* and the *L. kluyveri* species. Despite the absence of Mlh2 that restrains the extent of gene conversion tracts in *S. cerevisiae*, *L. waltii* gene conversion tract lengths are similar to those of *S. cerevisiae*. Remarkably, we found no conservation of the recombination hotspots across the *Lachancea* genus unlike what was observed in the *Saccharomyces* genus [4,21]. Finally, consistent with the loss of several ZMM proteins, the crossover interference signal in *L. waltii* is minimal and much reduced compared to *S. cerevisiae*.

## Results

### Genetic diversity across the *Lachancea waltii* species

To determine the meiotic recombination landscape of the *L. waltii* species, we first explored the genetic diversity of a collection of natural isolates in order to identify two parental polymorphic isolates to generate a diploid hybrid and its corresponding meiotic progeny. The collection used for resequencing consists of all the seven currently available isolates of *L. waltii*, including the CBS 6430 reference isolate. This strain was isolated in Japan from the tree *Ilex integra* and its genome was fully sequenced, assembled and annotated previously [46,62,63]. It consists of 10.2 Mb spreading over eight chromosomes (A to H). The six remaining *L. waltii* natural isolates come from either trees or insects from Canada (S2A Table). We resequenced the full genome of each isolate via a short-read Illumina sequencing strategy, generating a 75-fold genomic coverage on average. The reads associated with each sample were mapped against the CBS 6430 reference sequence. Variant calling allowed to detect a total of 954,071 SNPs distributed across 227,304 polymorphic sites (S3 Table). In addition, the reads coverage profiles showed the absence of aneuploidy in this set of strains and a single segmental duplication of approximately 130 kb was detected on chromosome A in the CBS 6430 strain (S1 Fig). Overall, we observed a genetic divergence of approximately 1.5% (or 160,000 SNPs) between the six strains isolated from Canada and the reference isolate, highlighting the presence of two subpopulations. The Canadian subpopulation shows a sequence divergence ranging from 0.31 to 0.65% (S2 Fig; S4 Table).

### Extensive LOH tracts in *L. waltii* hybrids

The *L. waltii* species propagates vegetatively as a haploid. Opposite mating types are seen, but not at the same frequency, diploids are rare, and sporulation is infrequent. We generated hybrids between the LA128 and LA136 strains as well as between the LA133 and LA136 isolates owing to their SNP density, approximately 0.64 and 0.4%, respectively. We tested several mating regimens, with different mating times and media (YPD, SC, Malt agar). Successful mating and subsequent euploid diploids were only achievable after 48 or 72 hours of mating on Malt agar. Importantly, in contrast to *S. cerevisiae*, mating frequencies in *L. waltii* are very low. In addition, *L. waltii* exclusively mates and sporulates on Malt agar and we detected hybrid sporulation as early as 12 hours after incubation on malt agar, albeit at a very low frequency, 1.9% (S3 Fig) [64]. We sequenced nine hybrids of the LA128/136 cross (four at 48h and five at 72h) and five hybrids of the LA133/136 cross (two at 48h and four at 72h).

The analysis of the genome of these hybrids highlighted frequent aneuploidies and a plethora of large regions of loss of heterozygosity (LOH) (S4 Fig). Aneuploidies of chromosomes B and D were prevalent in both hybrid backgrounds. Extra copies of chromosome B or D were observed five and eight times, respectively, among the nine LA128/136 hybrids. In the five

LA133/136 hybrids, extra copies of chromosome B or D were observed once and four times, respectively. Concerning LOH regions, the LA128/136 hybrids had a mean of 12 (±0.9) LOH tracts, and the LA133/136 hybrids had a mean of 21 (±2.1) LOH events. The average LOH event size was 149 kb (range: 22–376 kb). These events were further away from the centromeres (mean centromere distance- 412 kb) than the telomeres (mean telomere distance- 138 kb), with more than half of the LOH events coinciding with the last 20 kb of chromosome arms. Despite the many mating regimens tested, LOH regions were present in all the hybrids analyzed. Formation of LOH regions appear to be inevitable when generating *L. waltii* hybrids, although their origin remains unknown.

### *L. waltii* recombination population

LOH tracts and aneuploidies preclude optimal detection of recombination events. However, their presence in all the *L. waltii* hybrids led us to choose a LA128/136 hybrid that accumulated the least number of LOH tracts (n = 11) and that is aneuploid for chromosome B and D to make a segregant population and determine the *L. waltii* meiotic recombination landscape (Fig 1) [65]. The 11 LOH tracts encompass five out of the six euploid chromosomes and represent a total of 1.64 Mb. The LA128/LA136 hybrid contains 64,628 polymorphic sites distributed across the eight chromosomes. This corresponds to a divergence of ~0.64%, which is in the same order of magnitude as those of the hybrids used to study meiotic recombination landscapes in *S. cerevisiae* and *L. kluyveri* [53,61,66,67]. The LA128/LA136 hybrid shows ~30% sporulation efficiency after 72 hours on malt agar medium, with 73% spore viability and 42.4% of full viable tetrads (S5 Table).

We generated 768 segregants from 192 full viable tetrads. We genotyped this F1-offspring population by Illumina whole genome sequencing and identified recombination events based on 50,274 SNPs parental inheritance, disregarding chromosome B and D as well as the 11 LOH regions for the analysis of recombination. Overall, we generated a recombination map for 5.7 Mb out of the 10.2 Mb of the genome.

To validate the 50,274 SNPs used for genotyping, we computed their 2:2 segregation and their pairwise recombination fractions. This analysis revealed the presence of a reciprocal translocation between chromosomes A and F in the parental strains compared to the CBS 6430 reference genome (S5A Fig; 730 kb on chromosome A and 670 kb on chromosome F). This translocation could also be identified through genome assemblies and is shared by all the Canadian isolates (S5B Fig). Crossovers and noncrossovers detected around the translocation breakpoints on chromosomes A and F (1 kb window on chromosome A, 12.2 kb window on chromosome F empirically determined based on the range of misaligned reads around the breakpoints) were excluded from downstream analyses. Overall, the median distance between consecutive markers in euploid heterozygous regions of the LA128/136 hybrid is 159 bp. These markers are distributed rather evenly across the genome, giving a resolution comparable to previous studies in *S. cerevisiae* and *L. kluyveri* [53,61,66,67].

### *L. waltii* recombination landscape

In the 192 four-spore viable tetrads analyzed, a total of 4,049 crossovers and 1,459 noncrossovers were identified, leading to a median of 19 crossovers and 7 noncrossovers per meiosis (S6 Fig; S6 and S7 Tables). Additionally, a detectable gene conversion tract was observed in 73% of the crossovers. The crossover and noncrossover rates were 3.4 crossovers/Mb and 1.2 noncrossover/Mb, respectively. Based on this, we estimated nearly 34 crossovers and 12.5 noncrossovers per meiosis genome wide (including regions under LOH and chromosomes B and D). With an intercept around 1 (0.72), the average frequency of crossovers per

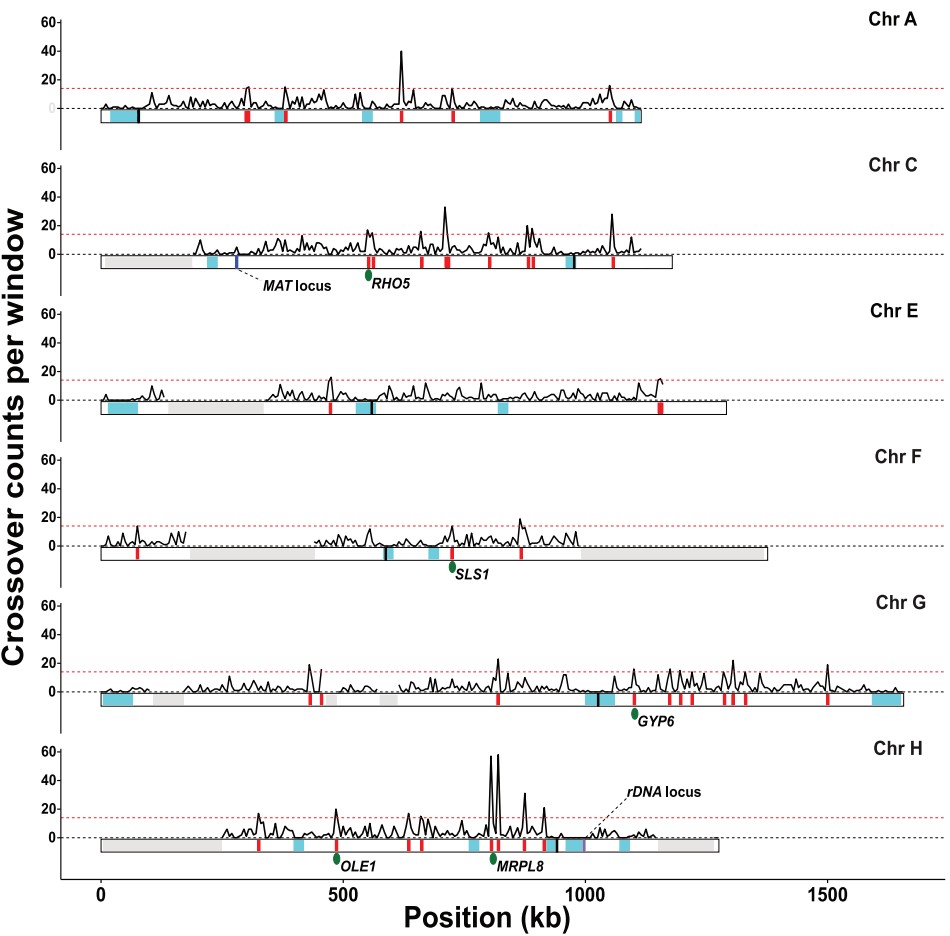

**Fig 1. *L. waltii* recombination map.** Density of crossovers along the genome using a 5 kb window. Grey shaded areas represent LOH regions from the parental hybrid. Horizontal dashed red line represents crossover hotspot significance threshold (see Materials and methods). Under the density plot, crossover coldspots and hotspots are shown in cyan and red, respectively. For each chromosome a black dash represents centromere position. Blue and purple dashes represent MAT locus on chromosome C and rDNA locus on chromosome H respectively. *S. cerevisiae* conserved hotspot positions are represented by green ovals [53,61]. Note that two COs frequently took place in the same meiosis in the two hotspots near *MRPL8* and involved only two chromatids most of the time.

chromosome exhibits a linear relationship with LOH corrected chromosome size (Fig 2A). By contrast to *S. cerevisiae*, crossover density has no significant inverse correlation with chromosome size in both *L. waltii* and *L. kluyveri* (Fig 2B). Similar results have been seen in crossover interference mutants in *S. cerevisiae* [33,68].

Most ancestral heterozygous SNPs (84.8%) segregated 2:2 in *L. waltii*. The median sizes of conversion tracts associated to crossovers (GCco) and noncrossovers were 2.0 kb and 1.8 kb, respectively, and were significantly different (Mann Whitney U test, p<0.0001; Fig 2C and S8 Table). Such GCco and noncrossover tract lengths are identical to those from *S. cerevisiae* but shorter than those from *L. kluyveri* (2.8 and 3.0 kb for GCco and noncrossovers, respectively: Mann Whitney U test, p<0.0001).

Approximately 3% of *S. cerevisiae* meioses exhibit chromosomes that segregate without crossover, *i.e.* non-exchange or $E_0$ chromosomes (0 for no exchange), but only small chromosomes are impacted [53,68]. In contrast, almost 45% of *L. kluyveri* meioses had an $E_0$ chromosome, disregarding chromosome C where crossovers are suppressed in a large region of the

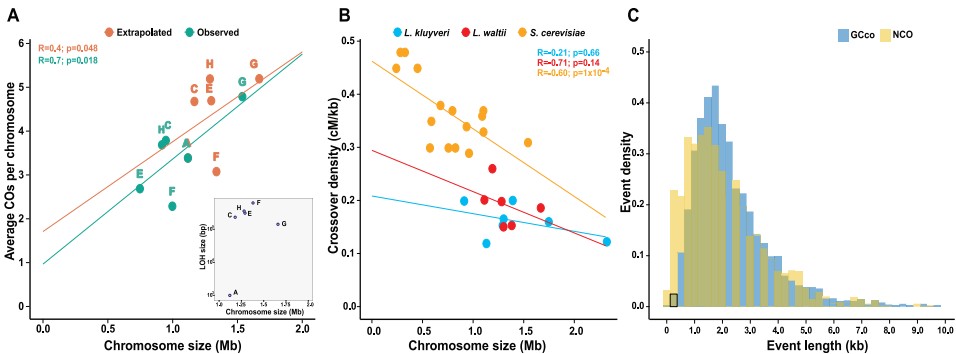

**Fig 2. *L. waltii* recombination events. A**. Frequency of crossovers in *L. waltii* are related to chromosome size. The 'observed' version corresponds to the average number of CO detected compared to the studied fraction of each chromosome, *i.e.*, excluding the LOH regions (intercept = 0.72, slope = $2.65 \times 10^{-6}$). The 'extrapolated' version represents the corrected number of COs relative to the actual chromosome size (intercept = 1.4, slope = $2 \times 10^{-6}$). The corrected CO number was estimated by extending the CO rate (3.4 CO/Mb) to the whole chromosome. [Inset: cumulative sizes of regions under LOH relative to the actual chromosome size.] **B**. Recombination rate (cM/kb) plotted against chromosome size for *L. waltii* with the extrapolated values (R = -0.71, p = 0.14), *L. kluyveri* (R = -0.21, p = 0.66) and *S. cerevisiae* (R = -0.6, p = 0.6, p = $1 \times 10^{-4}$). *L. kluyveri* data from [61], *S. cerevisiae* data pooled from [68] and [33]. **C**. Distribution of gene conversion tract sizes associated to crossovers (GCco) and noncrossovers.

left arm [61]. In *L. waltii*, the incidence of $E_0$ chromosomes is also high, with 22.9% of meioses displaying at least one and 8.3% exhibiting two or more $E_0$ chromosomes (Fig 3). This result comes with the caveat that LOH in the parental hybrid may exaggerate the $E_0$ estimates. However, *L. waltii* 1.12 Mb long chromosome A, which displayed no LOH events, was found to be $E_0$ in 5.2% of the meioses. This contrasts with *S. cerevisiae* where chromosomes of equivalent size never show zero crossover [26,53,68,69]. Overall, unlike *S. cerevisiae*, both *Lachancea* species seem to be able to segregate accurately a high frequency of large chromosomes without crossover. Interestingly, the frequency of $E_0$ chromosomes in *L. waltii* was lower than in *L. kluyveri*, despite lacking several pro-crossover ZMM genes.

## *L. waltii* recombination hotspots and coldspots

PRDM9 dependent DSB hotspots evolve rapidly compared to PRDM9 independent DSB hotspots that characterize many non-metazoans organisms. PRDM9 independent DSB hotspots mainly depend on DSB formation in functional elements such as gene promoters and were

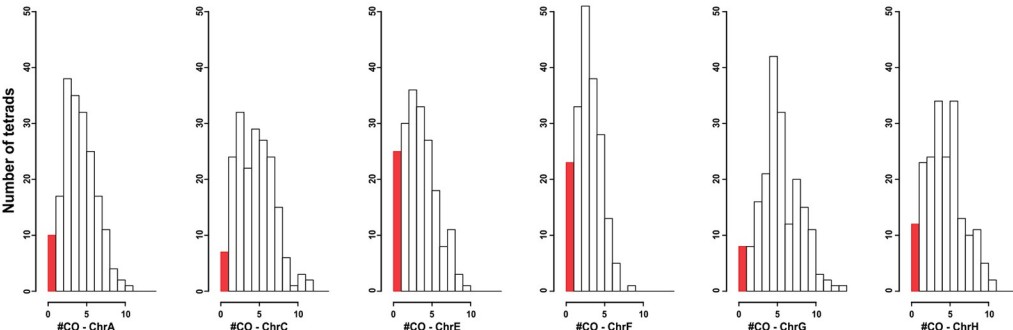

**Fig 3. Non-exchange chromosome ($E_0$) frequencies in *L. waltii*.** Histogram representing the frequency of crossovers per meiosis in each *L. waltii* chromosome. The red bars represent instances where no crossover was detected on the chromosome ($E_0$).

shown to be well conserved in positions and intensities across *Saccharomyces* species. This resulted in the concept of "nonparadoxical evolutionary stability of the recombination initiation landscape in yeast", mainly explained by the selective pressure on the underlying functional elements determining DSB hotspots [4]. However, this nonparadoxical evolutionary stability is not absolute and its extent is not known. So far, it is restricted to species that show an overall well conserved synteny, while many factors including sequence divergence and synteny rearrangements may affect DSB hotspot strength and location, for instance because of DSB hotspot competition [70,71] and DSB hotspot interference [72]. Because DSB hotspots determine crossover hotspots, evaluating the conservation of crossover hotspots between *S. cerevisiae* and *L. waltii* was important to see if the evolutionary stability of the recombination landscape extended beyond the *Saccharomyces* species.

To identify potential crossover hotspots or coldspots, we computed the local recombination rate using the 4,049 single crossovers (Fig 1). Recombination hotspots were identified using a window size of 5 kb and the significance was assessed using permutated datasets ($10^5$ permutations, FDR of 2%, see Materials and methods), similar to what we did for *L. kluyveri* [61]. Recombination coldspots were identified with the same approach using a 20 kb window size to increase detection power. Overall, we detected 21 crossover coldspots and 37 crossover hotspots that were represented by 20 crossovers on average per 5 kb window in *L. waltii* (Fig 1). Among the seven *L. kluyveri* crossover hotspots, *PIS1* and *PRM2* do not have annotated orthologs in *L. waltii* and *OST6* ortholog is located on *L. waltii* chromosome D which was discarded from our analysis because of the presence of an aneuploidy. None of the four remaining *L. kluyveri* hotspots (*ALT2*, *RAS1*, *GPI18* and *VMA10*) were detected as hotspots in *L. waltii*, suggesting a lack of crossover hotspot conservation between *L. kluyveri* and *L. waltii* [61].

To determine the conservation of crossover hotspots and coldspots between *L. waltii* and *S. cerevisiae*, we used our procedure to analyze *S. cerevisiae* recombination data from a large dataset gathering a total of 72 S288c/YJM789 hybrid *S. cerevisiae* meioses, that amassed a total of 6,948 crossovers [53,69,73]. Factoring the genome sizes of the two species over which crossover detection was made, the *L. waltii* dataset was substantially more crossover dense (710 crossovers per Mb) than the *S. cerevisiae* dataset (575 crossovers per Mb). Overall, we detected 32 coldspots and 65 hotspots in *S. cerevisiae* (S7 Fig). The *S. cerevisiae* hotspots were represented by an average of 15 crossovers per 5 kb window compared to *L. waltii* hotspots which were represented by an average of 20 crossovers per 5 kb window (p<0.01; Wilcoxon test). Importantly, 57 (87%) hotspots detected by our procedure were also within or adjacent to hotspots detected previously (Mancera et al. 2008) (S9A Table). This number increased to 75 hotspots when we relaxed our FDR to 5% (S9A Table). The fact that the previous study detected 92 crossover hotspots likely results from less stringent criteria. Next, we determined the conservation of the 92 *S. cerevisiae* hotspots reported previously [53]. This is a more arduous task as synteny between these species is poorly conserved. To do so, we retrieved conserved syntenic blocks in *L. waltii* genome containing at least two *S. cerevisiae* orthologs associated with one hotspot (see Materials and methods). Eventually, 86 partial syntenic blocks were found in *L. waltii* genome, 51 of which were located in the analyzed part of the genome. Among the 92 *S. cerevisiae* crossover hotspots, we identified the orthologous regions for 82 of them (S9A Table). Reciprocally, among the 37 *L. waltii* crossover hotspots, we identified the orthologous regions for 19 of them (S9B Table). Overall, *L. waltii* and *S. cerevisiae* share only the five following crossover hotspots *RHO5*, *SLS1*, *GYP6*, *OLE1* and *MRPL8*. This shows that crossover hotspots are overall not conserved between *L. waltii* and *S. cerevisiae* (Figs 1 and S7) [53,74].

As for *S. cerevisiae* and *L. kluyveri*, recombination coldspots in *L. waltii* include the eight centromeres, the surrounding of the rDNA locus and all LOH-free sub-telomeric regions (chromosomes A, E, G), except the chromosome F left sub-telomere [9,49,52,53,61,75,76].

Furthermore, nine recombination coldspots are located away from any centromere, subtelomere and the rDNA locus (Fig 1). Overall, 77 genes were identified in the coldspots. Interestingly, they include LAWA0A06634g and LAWA0E09516g, the orthologs of *DMC1* and *MEI5* which encode the meiotic specific recombinase and one of its accessory factors, respectively, and LAWA0F07800g, the ortholog of *SLX4* which encodes a scaffold protein that controls and coordinates the action of multiple structure specific endonucleases and interacts with other DNA repair factors. However, we did not find any functional class of genes to be significantly enriched in coldspots. Therefore, it is not obvious that the previous evidence showing that meiotic genes tend to be protected against meiotic DSBs in *S. cerevisiae* applies to *L. waltii* [9].

A striking feature of *L. kluyveri* recombination landscape was the absence of recombination on the entire left arm of the chromosome C (Lakl0C-left) [61]. The synteny of this large 1-MB region is conserved over 600 kb between *L. kluyveri* and *L. waltii*. By contrast to *L. kluyveri*, the *L. waltii* region orthologous to the Lakl0C-left exhibits a recombination rate (3.56 crossovers/Mb) similar to the rest of the genome (Fig 1).

### *L. waltii* crossovers exhibit interference

Crossover interference is an interactive process between crossovers along chromosomes, enabling non-random distribution of crossovers [77]. We analyzed crossover interference genome-wide using the coefficient of coincidence (CoC) method (Materials and methods and see [33]). CoC is the ratio of the observed frequency of crossovers in two consecutive chromosomal intervals out of the expected frequency of such double crossovers if they arose independently in the two intervals. *L. waltii* crossovers exhibit interference over only short distances (up to 25 kb), while in *S. cerevisiae* it is positive up to a 100 kb interval size (Fig 4A). Only chromosomes A, E, and H display interference over a 25-kb interval, but chromosomes C, F, and G do not (Fig 4B). Unfortunately, the available *L. kluyveri* crossover dataset did not allow such a CoC analysis to be performed due to the small sample size.

The strength of interference can also be determined by modelling inter-crossover distances using a gamma distribution: $\gamma = 1$ indicates no interference, whereas $\gamma > 1$ corresponds to positive interference. The $\gamma$-values of the gamma-fit distributions for *S. cerevisiae*, *L. kluyveri* and *L. waltii* are 1.96, 1.47 and 1.23, respectively (Figs 5 and S8) [61]. The gamma-fit distribution of inter-crossover distances in *L. waltii* is significantly different from both of *S. cerevisiae* and *L. kluyveri* as well as from a distribution of $\gamma = 1$ (K-S test, p<0.0005). Overall, the CoC method and the modelling with a gamma distribution support the presence of crossover interference in *L. waltii* with a reduced strength relative to *S. cerevisiae*.

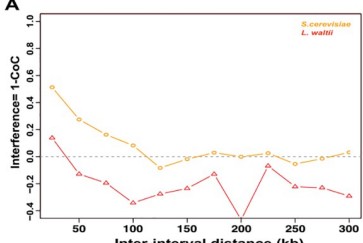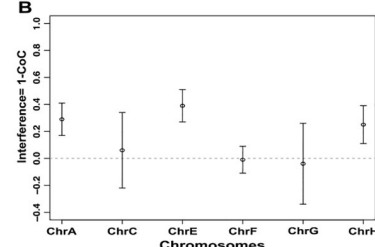

**Fig 4. Crossover interference in *L. waltii*.** **A**. Crossover interference (1-CoC) calculated for a bin size of 25 kb and inter-interval distance of 25 kb (i.e., for adjacent intervals) in *L. waltii* (triangle) and *S. cerevisiae* (circle). **B**. Chromosome-wide interference (1-CoC) in 192 *L. waltii* tetrads calculated for a bin size of 1/ (5* (mean inter crossover distance)) for adjacent bins using MAD patterns [96].

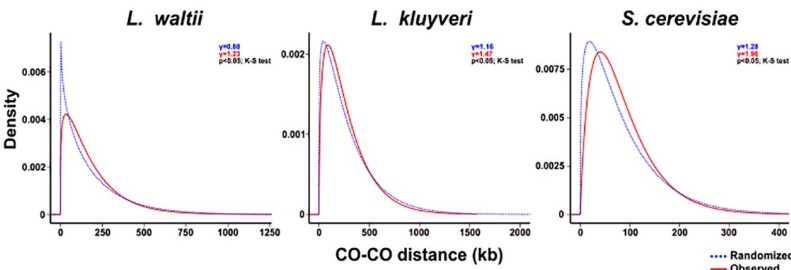

**Fig 5. Interference in *L. waltii*.** A gamma law fit of the distribution of distances between pairs of adjacent crossovers is represented compared to a randomized distribution of COs across tetrads. Gamma parameters were as follows: for *L. waltii*, γ = 1.23 and θ = 145.9, for *S. cerevisiae*, γ = 1.96, θ = 61.7 and for *L. kluyveri*, γ = 1.47, θ = 187. γ represents the shape parameter and θ represents the scale parameter. Distribution of inter-crossover distances depicted in S8 Fig.

We further modelled the crossover distribution in *L. waltii* with the CODA package, using the two-pathway model that incorporates contributions from both interfering and non-interfering crossover pathways [78]. This analysis suggests that only 7.7% of the crossovers in *L. waltii* are non-interfering, like *S. cerevisiae* with 5.8% (S10 Table). Altogether, our data suggests that most crossovers in *L. waltii* interfere, but interference barely extends beyond 25 kb.

## Discussion

### Mechanism of LOH formation: An under-estimate of the importance of RTG in every yeast species?

Despite our efforts to curb the time cells spent on malt agar, the media that permits *L. waltii* to mate and sporulate, we were unable to recover hybrids devoid of LOH (S4 Fig). Several pathways may have led to these LOH events (S9 Fig). LOH within *L. waltii* hybrids may have resulted from recombination events during mitotic propagation (S9A Fig). Since the mitotic propagation time was very limited, such events may have been promoted by leaky Spo11 activity. Alternatively, such events may have resulted from intra-tetrad mating (S9B Fig), and/or abortive meiosis often known as "return to growth (RTG)" (S9B Fig) [61,79,80]. The generation of LOH regions via any of these pathways is expected to be accompanied by a fraction of reciprocal crossovers in the ensuing hybrids, which should manifest as four-strand double crossovers in the corresponding tetrads [61]. In agreement with this, we identified five loci associated with four-strand double crossovers in most of the tetrads analyzed. Two of these loci that fall within LOH regions from the hybrid show evidence of additional CO that result in apparent single COs. Overall, without additional information, it is impossible to determine which of the three mechanisms of LOH formation took place.

The prevalence of LOH in *L. waltii* hybrids from two distinct genetic backgrounds led us to suspect that they may form in nature. By unraveling the expression of recessive alleles as well as new allelic interactions, LOH formation results in phenotypic variation and is therefore a driving force in genome evolution. LOH regions accumulate rapidly in response to both endogenous and exogenous stress [81,82]. As a result, in some yeast species including *L. waltii* and *L. kluyveri*, such LOH-generating processes could be stress adaption mechanisms, the generality of which is debatable.

### Which nuclease(s) make(s) crossovers in *L. waltii*?

So far, interfering crossovers were known to result from the ZMM pathway which Holliday junction resolvase activity comes from the action of the Mlh1-Mlh3 endonuclease optimized

by the presence of Exo1 (independently of its nuclease activity), Msh4-Msh5, RFC and PCNA [37,39,83]. Interfering crossovers are also known to form in the context of the synaptonemal complex. Zip2, Zip4 and Spo16 that form the ZZS complex promote interfering crossovers and connect both the chromosome axis and Holliday junctions (HJs) with high affinity, as well as the other interfering crossover promoting factor Zip3. The absence of the ZZS complex, Zip3 and Msh4-5 in *L. waltii* questions the involvement of Mlh1-3 in the formation of interfering crossovers. Either Mlh1-3 evolved to resolve HJ junctions in the absence of its accessory factors, or this activity has been taken over by other structure specific nucleases. In addition to the ZMM proteins, Mlh2 is also absent from *L. waltii*. In combination with Mlh1, it forms the MutLß complex that limits D-loop extension and likely the subsequent gene conversion tracts associated to both noncrossovers and crossovers in *S. cerevisiae* [84,85]. Interestingly, we found that the gene conversion tract lengths are of similar sizes in *L. waltii* and *S. cerevisiae*, and smaller than in *L. kluyveri*. This suggests that the MutLß role in limiting D-loop extension is dispensable in *L. waltii* to generate short conversion tracts, and hence that subtle differences exist in the biology of recombination intermediates between *L. waltii*, *L. kluyveri* and *S. cerevisiae*. These differences are likely at the root of the lower frequency of complex recombination events (classes c-f, S6 Fig) seen in *L. waltii* in comparison with *S. cerevisiae*, where they represent ca. 1.5% and 10% of all events, respectively [53].

## Is Mer3 involved in meiotic recombination in *L. waltii*?

While part of the ZMM dependent crossover pathway, Mer3 likely plays additional roles during meiotic recombination as suggested by the much higher Mer3 foci numbers compared to the observed crossovers numbers in different species [35]. In addition, in combination with Mlh1-Mlh2, Mer3 limits the extent of DNA repair synthesis during both crossovers and noncrossovers formation in *S. cerevisiae* [84,85]. Because most noncrossovers form independently of the ZMM pathway, this further supports that Mer3 functions extend outside the ZMM pathway. Both Mlh2 and the ZMM proteins Zip2, 3, 4, Spo16, Msh4 and Msh5 being absent in *L. waltii*, this raises the question of any Mer3 meiotic role in this species. Interestingly, we previously reported that Mer3 has been lost four independent times within the *Lachancea* clade, after the loss of Zip2, 3, 4, Spo16, Msh4, 5 and Mlh2 that occurred early, right after the divergence of *L. kluyveri* from the rest of the clade [46]. Eventually, Mer3 is still present in *L. waltii*, but absent from its closest relative *Lachancea thermotolerans*. One could speculate that these frequent losses of Mer3 along the *Lachancea* phylogeny result from a loss of selective pressure on Mer3. Although this could support an absence of meiotic function of Mer3 in *L. waltii*, this issue needs to be experimentally addressed.

## $E_0$ chromosomes

The high fraction of $E_0$ chromosomes in *L. waltii* contrasts with the low level observed in *S. cerevisiae*. This could come from the lack of recombination information from a large fraction of the genome corresponding to LOH regions in the staring hybrid. However, this argument can be discarded by chromosome A (and G to a lesser extent) that has no LOH and that still shows about 5% $E_0$ among the 192 tetrads. This high fraction of $E_0$ is like that observed in *L. kluyveri* chromosomes except for the mating type chromosome where the fraction of $E_0$ reached 50% likely because of the absence of recombination on its 1 Mb-long left arm. This supports the existence of an efficient distributive chromosome segregation mechanism in *L. waltii*, which would be more important than in *S. cerevisiae*, as postulated for *L. kluyveri* [61].

## Crossover interference in *L. waltii*

Although relatively weak compared to other species such as *C. elegans*, crossover interference exists in *S. cerevisiae* and has been shown to involve Top2 and Red1 through sumoylation [86]. Top2 and Red1 are conserved in *L. waltii*. However, Zip2, Zip3, Zip4, Spo16, Msh4 and Msh5, that likely play a role in crossover interference implementation as the corresponding mutants accumulate non-interfering crossovers in different species, have been lost in *L. waltii* [35]. Consequently, it was expected that *L. waltii* was defective for crossover interference. Interestingly, we found evidence of crossover interference in *L. waltii*, albeit with reduced strength compared to *S. cerevisiae*. Although the optimal metrics for crossover interference measure is meiotic chromosome axis length, crossover interference hardly extends beyond 25 kb in *L. waltii*. This size is in the range of size of a single chromatin loop in *S. cerevisiae* as first determined by electron microscopy and later on by HiC [87–89]. One explanation for this reduced but not null crossover interference could be that crossover interference is normally established but not properly implemented due to the absence of Zip2, Zip3, Zip4, Spo16, Msh4 and Msh5. Alternatively, it is possible that the crossover interference signal detected in *L. waltii* uniquely results from the non-uniform distribution of meiotic DSBs. Indeed, previous studies already reported weak but significant interference between crossover precursors both in tomato and in mouse [25,90,91]. In addition, a modeling study also supported an even patterning of meiotic DSBs in *S. cerevisiae*, and further showed that even DSB patterning *per se* was enough to generate a positive CoC signal [83–85]. Overall, the crossover interference signal detected in *L. waltii* is compatible with a signal solely resulting from an even distribution of meiotic DSBs. It supports the already formulated view that crossover interference is multilayered, with a short-range component influenced by positive interference between DSBs and at least a long-range component [25]. Positive DSB interference has been shown to rely on Tel1 in *S. cerevisiae* [33,72]. The *L. waltii* context supports that a long-range component controlling crossover interference involves some or all Zip2, Zip3, Zip4, Spo16, Msh4 and Msh5 in *S. cerevisiae*. In contrast to yeasts, another layer of regulation exists at least in *A. thaliana* and *C. elegans* where the central element of the synaptonemal complex, where polymerization is independent of ZMM action, was shown to play a role in establishing strong crossover interference [27–29].

## Crossover hotspots conservation

DSB hotspots, that determine the location of recombination hotspots in many species including *S. cerevisiae* and likely any budding yeast, and recombination hotspots, are relatively stable across *Saccharomyces* species [4,21]. Conservation of recombination hotspots has also been observed across finch species, where DSB hotspots are also determined simply by nucleosome free regions and where PRDM9 is absent [10]. Such a situation contrasts with the lack of recombination hotspots conservation we observed here between *L. kluyveri* and *L. waltii*, two *Lachancea* species that diverged about 80 million years ago. A possible explanation is that the genomes of the *Saccharomyces* yeasts analyzed are mostly colinear, with very limited genomic rearrangements, while about 130 chromosome translocations and inversions have been inferred between the extent *L. kluyveri* and *L. waltii* species [46]. Although we postulated a similar argument for the lack of conservation of recombination hotspot between *L. kluyveri* and *S. cerevisiae*, the present study illustrates that the loss of recombination hotspots can occur at a much-reduced evolutionary time such as the one between *L. kluyveri* and *L. waltii*.

## Materials and methods

### Yeast strains and growth conditions

The *Lachancea waltii* natural isolates and the modified strains used in this study are described in S1A and S1B Table. Components for cell culture media were purchased from MP biomedicals. Strains were grown in standard YPD medium supplemented with G418 (200 μg/ml) or nourseothricin (100 μg/ml) and agar 20 g/l for solid medium at 30°C. Mating and sporulation were performed on DYM medium (yeast extract 0.3 g/l, malt extract 0.3 g/l, peptones 0.5 g/l, dextrose 1 g/l and agar 20 g/l) at 22°C.

### Generation of parental strains

Stable haploid parental strains were obtained by replacing the *HIS3* locus with G418 or nourseothricin resistance markers. The *his3Δ::KanMX* cassette was amplified from the strain 78 using primer pairs HIS3-F/HIS3-R. The *his3Δ::NatMX* locus was amplified by a two-steps fusion PCR: the flanking regions of the *HIS3* locus were amplified from the strain LA128 using HIS3-F/HIS3-fusionTermF and HIS3-R/HIS3-fusionProm-R while the *NatMX* cassette was amplified from the plasmid pAG36 using HIS3Fusion TEFProm-F/HIS3Fusion TERterm-R. pAG36 was a gift from John McCusker (Addgene plasmid# 35126) [92].

The transformation of parental strains with the *NatMX* or *KanMX* cassettes was performed by electroporation as described by using 1 μg of DNA and electroporation at 1.5 kV, 25 mF and 200 Ω using a GenePulser (Biorad) [64]. To confirm successful replacement of the *HIS3* locus, colony PCR were performed using primer pairs HIS3-F/Kan-R or HIS3-F/Nat-R. All primers used are detailed in S11 Table.

### Mating, sporulation, and spore isolation

For mating, LA128 (*his3Δ::KanMX)* and LA136 (*his3Δ::NatMX)* were mixed on DYM plates for 72 h at 22°C. Double resistant cells to G418 and nourseothricin were selected on YPD-G418-Nourseothericin plates. Single colonies were purified by streaking on YPD-G418--Nourseothericin plates. The diploidy in hybrids LA128 (*his3Δ::KanMX*) / LA136 (*his3Δ::NatMX*) was verified by flow cytometry.

After ploidy validation, one hybrid was selected and sporulated for 2–3 days on DYM plates at 22°C. Tetrads dissections were performed using the SporePlay (Singer Instrument) without any pre-treatment. Dissection of about 1,000 tetrads showed 2:2 segregation of the *KanMX* and *NatMX* markers.

### Genomic DNA extraction

Total genomic DNA of the *L. waltii* natural isolates as well as the generated hybrids were extracted using a modified MasterPure Yeast DNA purification protocol (Lucigen).

Total genomic DNA of the 768 segregants was extracted using the 96-well E-Z 96 Tissue DNA kit (Omega) following a modified bacterial protocol. Cells were grown overnight at 30°C with agitation at 200 rpm in 1 ml of YPD in 2 ml 96 deep square well plates, sealed with Breath-Easy gas-permeable membranes (Sigma-Aldrich). Cells were centrifuged for 5 min at 3,700 rpm and the cell wall was digested for 2 h at 37 °C in 800 ml of buffer Y1 (182.2 g of sorbitol, 200 ml of EDTA 0.5 M pH 8, 1 ml of beta-mercaptoethanol in a total of 1 l of $H_2O$) containing 0.5 mg of Zymolase 20T. The cells were pelleted, resuspended in 225 ml of TL buffer containing OB protease and incubated overnight at 56 °C. Subsequently, DNA extraction was continued following the manufacturers protocol.

DNA concentration was measured using the Qubit dsDNA HS assay (ThermoFischer) and the fluorescent plate-reader TECAN Infinite Pro200 and DNA quality was evaluated using a NanoDrop 1000 Spectrophotometer (ThermoFischer).

## Illumina high-throughput sequencing

For the seven *L. waltii* isolates, genomic Illumina sequencing libraries were prepared with a mean insert size of 280 bp and subjected to paired-end sequencing (2×100 bp) on Illumina HiSeq 2500 sequencers by the BGI.

For the hybrids and the 768 segregants, DNA libraries were prepared from 5 ng of total genomic DNA using the NEBNext Ultra II FS DNA Library kit for Illumina (New England Biolabs). All volumes specified in the manufacturer's protocol were divided by four. The adaptor-ligated DNA fragments of about 300-bp were amplified with 8 cycles of PCR using indexed primers. A combination of 48 i7 oligos (NEBNext Multiplex Oligos for Illumina, NEB) and 24 i5 oligos (Microsynth) were designed enabling multiplexing up to 1152-samples. After quality check using a Bioanalyzer 2100 (Agilent Technologies) and quantification using the Qubit dsDNA HS assay, 4 nM of each of the 782 libraries were pooled and run on a NextSeq 500 sequencer with paired-end 75 bp reads by the EMBL's Genomics Core Facility (Heidelberg, Germany).

## Mapping and Single Nucleotide Polymorphisms (SNPs) calling

Sequencing reads were mapped to the *L. waltii* reference genome (obtained from the GRYC website (http://gryc.inra.fr/index.php?page=download) using bwa mem (v0.7.17). Resulting bam files were sorted and indexed using SAMtools (v1.9). Duplicated reads were marked, and sample names were assigned using Picard (v2.18.14). GATK (v3.7.0) was used to realign remaining reads. Candidate variants were then called using GATK UnifiedGenotyper.

## Segregation analysis

After variant calling, SNPs called in the LA128 and LA136 parents were first filtered (bcftools view, v1.9) to define a set of confident markers, corresponding to positions with a single alternate allele, supported by at least ten sequencing reads in each parent and with >90% of the sequencing reads covering either the reference or alternate allele. For each segregant resulting from the LA128 and LA136 cross, SNPs located at marker positions were extracted, and parental origin was assigned based on SNP correspondence between parents and spores at those positions.

To validate these SNPs as markers to generate the recombination map, their segregation among the progeny was investigated. If most of the markers (67%) follow the expected 2:2 mendelian segregation, a significant amount displays other patterns. Indeed, 20% of the markers show 0:4 / 4:0 segregation illustrating loss of heterozygosity (LOH) events in the LA128/136 hybrid. Distribution of these 0:4 / 4:0 SNPs along the genome shows that at least 1 LOH event occurred within 5 of the 8 chromosomes. In total LOH events represent 1.965 Mb of the 10.2 Mb total genome size. These LOH segments cannot be used for recombination studies and therefore have been discarded for the following analysis.

Another significant amount of SNPs encompassing all chromosomes B and D are deviating from 2:2 segregation with a more complex pattern. By looking at the coverage along the genome in LA128/136 parental hybrid, an aneuploidy with a supplemental copy of chromosome B and D was identified, explaining deviation from 2:2 segregation. Therefore, this aneuploidy was inherited in some of the segregants showing heterozygosity for theses chromosomes. In order to keep only euploid information for recombination analysis, we did not

consider chromosomes B and D in our analysis. In total, the combination of chromosomes B and D and LOH regions represents 4.5 Mb that were not considered for the recombination analysis that eventually encompassed 5.7 Mb.

## Whole genome assemblies

We used Abyss (v2.0.2) with the option '-k 80' to produce de novo assemblies for all the natural isolates. For each assembly, the scaffolds corresponding to the regions of interest (chromosome A from position 725,000 to 735,000 and chromosome F from 670,000 to 690,000) were detected through blastn similarity searches and compared to the reference genome with nucmer and mummerplot [93,94].

## Identification of recombination events

For each tetrad resulting from the cross between the LA128 and LA136 isolates, SNPs located at aforementioned marker positions were extracted, and parental origin was assigned based on SNP correspondence between parents and spores at those positions. The result was formatted as a segfile and used as input of the CrossOver python script (Recombine suite, [95]), using default parameters and adapted chromosome sizes/coordinates to fit the reference genome of *L. waltii* (File S1).

A large amount of LOH regions were present in the parental hybrid and was passed to the studied tetrads, introducing noise in the results obtained with CrossOver. In addition, chromosomes B and D, which are prone to aneuploidies, could not reliably be analyzed for the presence of crossovers and noncrossovers. Finally, a translocation event between chromosome A and chromosome F also introduced false positive crossing-over calls at the translocation breakpoints. Thus, crossover and noncrossover events reported in chromosomes B and D (approx. 2.35 Mb), in LOH tracts (mean size: 2.17 Mb per tetrad) and at the chromosomes A-F translocation breakpoints (13.2 kb) were masked. The remaining studied genome size after masking was of 5.7 Mb.

Note that the four tetrads 47, 51, 95 and 112 escaped our attention. They were included in the global analysis of the 192 tetrads while each of them contains a spore with a genotype incompatible with the three others.

## Determination of recombination hotspots and coldspots and conservation analysis

The distribution of the crossovers along the genome was assessed to identify recombination hotspots and recombination coldspots. Number of crossovers in 5 kb windows and 20 kb windows was computed to identify hotspots and coldspots, respectively. Significance thresholds indicating more or less crossovers in a window than expected by chance was determined using permutation test. The density of crossovers per window was simulated using $10^5$ random distributions of 4,049 crossovers following a binomial law with equal chances for each event to fall into an interval. For each of the $10^5$ simulations, the maximum or minimum number of crossovers per interval was extracted and the 2000th highest or lowest value set as the threshold to define hotspots and coldspots, respectively. Therefore, significance threshold for hotspots is set as >13 crossovers per window and <3 crossovers per window for coldspots.

Synteny between the 92 *S. cerevisiae* hotspots described by Mancera *et al.* 2008, and *L. waltii* was investigated. Each *S. cerevisiae* hotspot was associated with the five closest genes and their potential orthologs have been detected in *L. waltii* genomes using annotated genome of reference strain CBS 6430 [46,63]. When more than two orthologs linked to same hotspots were found in the same region (separated by less than 10 kb), the synteny was considered as

conserved and suitable for hotspot comparison. Hotspots were considered as conserved if they were separated by less than 5 kb.

## Evaluation of crossover interference

A gamma law was fitted to the distribution of all inter-crossover distances in the 192 tetrads using the fitdistr R function (R, MASS package). Then the shape (γ) and scale (θ) parameters of the fitted function were extracted. Crossover interference was evaluated by running a Kolmogorov-Smirnov test (using the ks.test function in R) between the distribution of all inter-crossover distances in the 192 tetrads and a gamma law of shape 1 (*i.e.*, without interference) and scale like the one previously determined. To assess interference by distance as pictured in Fig 4A, the CalculateInterference.py script from the Recombine suite was used, with parameters adapted to the experiment and genome of *L. waltii*, using an interval length of 25 kb [33].

CODA (V1.2) was used to assess interference and distinguish class I crossovers (interference dependent) from class II crossovers (interference independent) [78]. Relative positions of crossover events across each chromosome were computed and used as input for CODA. The gamma-sprinkling model was used to identify the two classes, using the "two-pathway" option with default min, max and precision parameters. The projected score was selected as the score type to fit the model, with $10^6$ simulations as suggested in CODA's help document. The hill-climbing algorithm was used as the algorithm of choice for determining optimum parameters.

## Supporting information

**S1 Table. Status of ZMM proteins in *L. waltii* in reference to *S. cerevisiae* and *L. kluyveri*.**
(XLSX)

**S2 Table. Strains in the study.** A. Natural *L. waltii* isolates collection. B. Strains generated and used in this study (Highlighted hybrid used to generate the mapping population).
(XLSX)

**S3 Table. Single nucleotide polymorphisms (SNPs) detected in the natural *L. waltii* isolates.**
(XLSX)

**S4 Table. Pairwise divergence (%) between the *L. waltii* isolates, based on SNPs.**
(XLSX)

**S5 Table. Meiotic spore viability in the LA128/LA136 hybrid.**
(XLSX)

**S6 Table. Summary of detected events in *L. waltii*, *S. cerevisiae* and *L. kluyveri*.**
(XLSX)

**S7 Table. Crossover events detected in the 192 *L. waltii* tetrads (ReCombine output, Anderson et al. 2011).**
(XLSX)

**S8 Table. Simple gene conversion (NCO) and crossover associated conversion (GCco) tracts detected across the 192 tetrads (ReCombine output, Anderson et al. 2011).**
(XLSX)

**S9 Table.** A. Table depicts the overlap of hotspots described by Mancera et al. 2008, without hotspot detection method at FDR 2% and FDR 5% respectively. Synteny in *L. waltii* highlighted. B. Table depicts the *L. waltii* hotspots with syntenic blocks in *S. cerevisiae*.

Conserved hotspots highlighted.
(XLSX)

**S10 Table. CODA output for *L. waltii* and *S. cerevisiae*.** Nu is the shape parameter of the gamma law, with nu< = 1 indicating the absence of interference, p is the proportion of type II crossovers (interference independent).
(XLSX)

**S11 Table. List of primers used in this study.**
(XLSX)

**S1 Fig. Sequencing coverage across the 8 chromosomes of the CBS6430 (LA126) reference isolate.** A large 130 kb segmental duplication is detected at the beginning of the chromosome A.
(TIF)

**S2 Fig. Phylogenetic relationship between the studied isolates, based on 227,304 polymorphic positions.** This tree underscores the presence of two main subpopulations in *L. waltii*, with LA126 lying away from the six other isolates.
(TIF)

**S3 Fig. Sporulation efficiencies (% cells MI + MII) in the LA128/136 and LA133/136 hybrids, post transfer to malt agar plates.** At least 300 cells were counted across three replicates.
(TIF)

**S4 Fig. Distributions of LOH regions across the independent diploids generated from the LA128/136 (n = 8, excludes the starting hybrid diploid of the mapping population) and LA133/136 (n = 5) hybrids.** The diploids were isolated at either 48 hours or 72 hours post mating from Malt-agar plates. LOH regions (defined as Peter et al. 2018) identified on the basis of the number of heterozygous sites per 50 kb windows are represented as vertical red columns.
(TIF)

**S5 Fig. Reciprocal translocation between chromosomes A and F in LA128 and LA136 compared to the reference strain.** A. Pairwise linkage disequilibrium for a representative subset of 5,542 markers. The markers in x-axis and y-axis are ordered according to their position along the reference genome and chromosomes are numbered from 1 to 8. Yellow represents high linkage disequilibrium and dark blue low linkage disequilibrium. Yellow spots between markers of the first and sixth chromosome highlight the presence of a translocation in the progeny compared to the reference genome. B. Translocation between chromosomes A and F was detected in all Canadian isolates (728, 798 on Chromosome A and 675, 008 on Chromosome F).
(TIF)

**S6 Fig. Representative allelic segregation patterns observed in the four spores of a tetrad, as defined in Anderson et al. 2011.** CO-Crossover, NCO-Non crossover, GC- Gene conversion, LOH-Loss of heterozygosity event during meiosis. Reported numbers are the counts of such events detected across the 192 tetrads. a) CO without GC. b-d) CO with GC on a chromatid either involved or not involved in the CO. e) CO with complex GC. f) CO with GC on a chromatid not involved in the CO. g) Double CO without GC. Note that five loci showed a double CO signature in all the tetrads. These double COs were inferred to have resulted from mitotic COs in the parental hybrid and were filtered out and not included here. However,

because of marker densities issues, 48 of the 68 double COs from this "g" category escaped the filtering. It remains a maximum of 20 double COs of this "g" class that are of meiotic origin. h) Double CO with GC. i) Double CO with complex GC. j) Simple NCO. k) 4:0 GC tract.
(TIF)

**S7 Fig. Density of crossovers along the genome using a 5 kb window in the *S. cerevisiae* genome (Mancera et al. 2008; Oke et al. 2014; Krishnaprasad et al. 2015 combined dataset).** Horizontal dotted red line represents crossover hotspot significance threshold (FDR 2%). Horizontal dotted green line represents crossover hotspot significance threshold (FDR 5%). Solid spheres represent (Green- orthologs in L. waltii; Red- No orthologs in L. waltii) Mancera et al. 2008 hotspots. Black diamond depicts the conserved hotspot in the two species.
(TIF)

**S8 Fig. Histograms representing the distributions of inter-crossover distances in actual and randomized tetrads.** For randomization, each CO is given a random tetrad number in the range of the experiment, then distances are calculated.
(TIF)

**S9 Fig. Possible scenarios leading to LOH outcomes in the *L. waltii* segregants.**
(TIF)

## Acknowledgments

We are grateful to Anne Villeneuve for fruitful discussions and her invaluable advice.

## Author Contributions

**Conceptualization:** Bertrand Llorente, Joseph Schacherer.

**Data curation:** Fabien Dutreux.

**Formal analysis:** Fabien Dutreux, Abhishek Dutta, Emilien Peltier, Anne Friedrich, Bertrand Llorente, Joseph Schacherer.

**Investigation:** Abhishek Dutta, Sabrina Bibi-Triki.

**Resources:** Bertrand Llorente, Joseph Schacherer.

**Supervision:** Joseph Schacherer.

**Writing – original draft:** Fabien Dutreux, Abhishek Dutta, Emilien Peltier, Anne Friedrich, Bertrand Llorente, Joseph Schacherer.

**Writing – review & editing:** Abhishek Dutta, Bertrand Llorente, Joseph Schacherer.

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
