## [Decision Letter · Decision Letter 0]

27 Jul 2022

Dear Joseph and Bertrand,

Thank you very much for submitting your Research Article entitled 'Lessons from the meiotic recombination landscape of the ZMM deficient budding yeast Lachancea waltii' to PLOS Genetics.

The manuscript was fully evaluated at the editorial level and by independent peer reviewers. The reviewers appreciated the attention to an important topic but identified some concerns that we ask you address in a revised manuscript

We therefore ask you to modify the manuscript according to the review recommendations. Your revisions should address the specific points made by each reviewer. In addition, you should address the points made below by myself (ML).

You will note that reviewer 2 has suggested analysis of several mutants, in particular mer3 and possibly mlh3 and mus81, at least to the level of spore viability/sporulation efficiency. I agree that these studies, in particular of MER3, would be important and would considerably add to the interest of the manuscript. However, I agree with reviewer 1 that the manuscript is of sufficient interest to not require this additional work, but again I think that the conservation of Mer3 is very interesting and a preliminary characterization of mer3∆ mutants, at least at the level of spore viability, would be quite interesting.

My other comments are as follows:

1. Fig 1, what are the CO frequency units?

2. Page 5, 13 Since Mlh2 appears to be absent but conversion tracts are cerevisiae-like, is there a functional Pif1 homolog in waltii, since Pif1 is required for the longer GC tracts seen in cerevisiae mlh2 mutants.

3. Page 5. Suggest “Finally, consistent with the loss of…”

4. Page 8. The Wilcoxon test is a paired-values test and is not appropriate here, as there are unequal numbers of GCco and NCOs and they are not paired with each other. A Mann-Whitney test, which is appropriate, reports that the GCco and NCO distributions ARE significantly different (p<0.0001). Please note that both Wilcoxon and Mann-Whitney are tests of the entire distribution, not the median. It is possible for medians to be the same while distributions are different.

5. Page 8. Based on the frequency of E0s, if extrapolated to all chromosomes, what is the expected spore viability assuming no distributive distribution? 

6. Page 10. Suggest “L. waltii crossovers exhibit limited interference” or "L. waltii crossovers exhibit interference over only short distances"

7. Page 11, figure 3, discussion. The negative interference displayed at all distances greater than 25 kb is consistent with selection at the cellular level for crossovers. Is it possible that this was imposed by selecting for 4 spore-viable tetrads? 

8. Page 11. “a gamma law of shape = 1”. Do you mean that k = 1? Might be worth briefly explaining what k and theta are in the legend to Figure 3.

9. References. Please make title capitalization style consistent, also italicize gene and species names.

We hope to receive your revised manuscript within the next 30 days. If you anticipate any delay in its return, especially if you decide to make and examine mer3∆ mutants, please let us know the expected resubmission date by email to plosgenetics@plos.org.

[LINK]

Yours sincerely,

Michael Lichten, Ph.D.

Academic Editor

PLOS Genetics

Kirsten Bomblies

Section Editor: Evolution

PLOS Genetics

Reviewer's Responses to Questions

**Comments to the Authors:**

Reviewer #1: Overview.

The manuscript describes the genome-wide analysis of meiotic recombination via the sequencing of haplotypes arising in meiotic spores of an intra-specific hybrid of the relatively unexplored yeast species, Lachancea waltii. The dataset is extensive (192 meioses), although detailed analysis of the results is relatively limited—in part because there are limited other data with which to compare results with. Crossover (CO) and noncrossover (NCO) frequency are calculated, CO distributions are assessed, and the evolutionary conservation of recombination hotspot positions is discussed. The results are compared to observations made in two other yeast species (S. cerevisiae, and L. kluyveri), the former of which many meiotic experiments have been performed, and much mechanistic detail has been inferred.

Overall, the study is somewhat limited in the impact that this dataset can make, but the study represents a commendable foray into the exploration of meiotic recombination outside of the classic species S. cerevisiae and S. pombe. I have only a few comments that might help the authors to improve the understanding and impact of the study.

1. General. I found the overall description of the LOH section (and the 4-strand double CO part), and in particular the parts referring to fig S3 (and Fig S3 itself), unclear. Perhaps break up into shorter paragraphs and add some helpful cartoon figures to explain the concepts/interpretations?

2. For the parts about hotspot conservation, it would help if the general concept here was more clearly introduced and explained. Is it expected that hotspots will be conserved at the location of orthologous genes? If so, by what mechanism is the gene (not the gene function) thought to be regulating this?

3. Fig 2. A&B. Please add R values and P value statistics to the linear correlations

4. Fig 2C. It would be helpful to replot these distributions normalised as a fraction of events so that the CO and NCO length distributions can be more easily compared.

5. Fig 3A. Is there a reason that the CoC curve is inverted compared to how it is has become to be plotted in the field (i.e. N. Kleckner)?

What does it mean that the L. waltii data show negative interference at all distances? Unless I missed it, this was not discussed, but seems of potential interest.

6. The authors suggest that short range positive interference may be due to DSB interference. If so, it may be expected that NCOs also show the same trend. Is the dataset deep enough to assess CoC between NCOs? It seems that with 192 meiosis it might be possible NCO distributions in a little depth (and perhaps CO-NCO relationships?).

7. Fig 3C. I’m not sure it makes sense to overlap these curves given that: A) the CO frequency varies greatly between each strain, and thus the CO-CO distribution will vary even if interference does not. B) If axis length is the relevant measure (of interference), without knowing the relative axis lengths between the different species, it is not clear how one would interpret the plot. Perhaps this point can be better highlighted/acknowledged?

Could each dataset be compared to a random (expected) distribution to determine by how much each CO-CO distribution deviates from random?

Stating the gamma shape parameters in text and figure here would also help.

8. Fig S6: Were there no events that had greater complexity that these examples? Is that surprising and/or consistent with data from S. cerevisiae?

9. Fig S7: Minor: Suggest to replot with chromosomes ordered either by size or by chromosome number.

10. Minor. The text refers to the concept that CO interference in S. cerevisiae “relies” on Top2 and Red1. This seems to overstate the observations of this prior study where mutations in those factors only reduced but did not abolish CO interference.

11. Minor: When describing loop sizes in S. cerevisiae, as well as the recent Hi-C estimates, it would make sense to also refer to much earlier EM estimates such as Moens & Pearlman 1998 etc.

Reviewer #2: In this article, the authors describe the meiotic recombination profile of a budding yeast species, Lachancea waltii, that diverged from S. cerevisiae before whole genome duplication, and lacks most of the ZMM genes involved in meiotic crossover formation, contrary to S. cerevisiae and another closely related species, Lachancea kluyveri that they studied previously.

For this, they established the genomic diversity of several waltii isolates, and chose to analyze two hybrid diploids that differ at many polymorphic sites, which allows monitoring meiotic recombination events. In addition to this genomic characterization, they establish the proper conditions to trigger sporulation, and notice that for unexplained reasons, the parental diploids already exhibit a high number of loss of heterozygocities (LOH) events, precluding the analysis of a part of the genome (about half).

Nevertheless, they managed to analyze the meiotic segregation profile of 192 tetrads, which they compare to that of budding yeast S. cerevisiae, and the “ZMM-proficient” L. kluyveri that the analyzed previously.

Their main conclusions are that like kluyveri, waltii exhibits less crossovers per genome than S. cerevisiae., that crossover density does not inversely correlate with chromosome size, contrary to S. cerevisiae, and that, unexpectedly, crossovers still show interference, despite the strain missing most ZMM genes. In addition, the crossovers hotspots that they determined thanks their high number of tetrads analyzed, do not match with those of S. cerevisiae, even in large synthenic blocs.

Globally, the paper is well written, the genomic analyses are well performed with sufficient statistical power due to the large number of tetrads analyzed, and the appropriate controls are provided.

However, it is mainly descriptive, and does not provide any experiment to support the proposed explanations to the observed recombination properties.

Major comments:

- Since MER3 and the SC gene ZIP1 are the only remaining ZMM genes in the waltii species, I think an essential experiment, which should be relatively straightforward, would be to perform a MER3 deletion, and at least analyze the spore viability and sporulation efficiency, to determine if, on its own, Mer3 is at all involved in meiotic recombination in this species. This is especially relevant since the authors mention that recombination tracts are of a similar size as those of S. cerevisiae, despite having lost the MLH2 gene. The Mlh1-Mlh2 heterodimer was described in budding yeast to limit the size of recombination events through it recruitment by Mer3. If meiotic recombination in waltii proceeds in a zmm-independent way, involving structure-specific nucleases such as Msu81-Mms4, Mlh2 would not be expected to influence the length of meiotic recombination events. It is therefore important to determine if Mer3 has any role in meiotic recombination on waltii. Also, a recent preprint on Biorxiv described that in addition to interacting with Mlh1-Mlh2 and limiting DNA synthesis, Mer3 may antagonize the action of the Sgs1 helicase to ensure interfering crossover formation (Altmannova et al). It may therefore participate, even as the sole ZMM, to promote some interfering crossovers.

- Another similarly very important experiment would be to delete MLH3, to determine if it is involved in meiotic recombination in waltii, like in species that have a functional ZMM upstream of Mlh1-Mlh3 endonuclease recruitment. In the same idea, MUS81 would also be interesting to delete, although it may induce a strong reduction of sporulation, as seen in cerevisiae even though it has only a minor role for crossover frequencies in this species.

Other comments:

- Page 4 Introduction, line 5: in S. cerevisiae, crossover interference IS (and not seems) independent of SC formation. In addition, here the papers by Pyatnitskaya et al (2022 Genes and Dev) as well as Amy Mc Queen (Voelkel-Meiman et al. 2015, 2016) on the remaining interference in SC central element deficient strains should be cited.

- Page 6, 2nd paragraph: Figures S3 and S4 numbers should be inverted, since current S4 is cited before current S3.

- Page 8: for clarity, the authors should better explain that the infer a total of 34 Cos and 12.5 NCO per meiosis although they assessed only 19 and 7, because they did not analyze a whole part of the genome (LOH and chromosomes B and D).

- Page 10, line 8: refer to Figure 1 at the end of the sentence. Same at the end of this chapter (after “the rest of the genome”).

-Page 14, Discussion: it should be mentioned that in A. thaliana and C. elegans, the SC polymerization is NOT dependent on ZMM proteins, which may explain why the SC seems to exert a strong effect on CO interference in these species, contrary to budding yeast S. cerevisiae.

- Page 21 legend of Figure 2B: typo: I think p=0.6 should be removed.

- Figure 2B: for clarity, in addition to figure legend, pvalues should be added on the graph, for the significant (S. cerevisiae) or not significant (kluyveri and waltii) inverse relationship between CO density and chromosome size.

**Have all data underlying the figures and results presented in the manuscript been provided?**

Reviewer #1: Yes

Reviewer #2: Yes

PLOS authors have the option to publish the peer review history of their article (what does this mean?). If published, this will include your full peer review and any attached files.

Reviewer #1: No

Reviewer #2: No

---

## [Decision Letter · Decision Letter 1]

30 Oct 2022

Dear Joseph,

With apologies for the delay, thank you very much for submitting your Research Article entitled 'Lessons from the meiotic recombination landscape of the ZMM deficient budding yeast Lachancea waltii' to PLOS Genetics.

The manuscript was fully evaluated at the editorial level and by independent peer reviewers. The reviewers were in large satisfied with the revision, but both raised points that could be addressed by additional text in a revised manuscript. You should consider this an opportunity that is not obligatory, as addressing these points will add some additional depth to the paper, but I will not be sending the revision out for further review.

Two technical points that should be addressed in this revision. First, the formatting of the references is inconsistent, as one would expect for the direct output of a reference manager program. Please make sure that journals are properly abbreviated, that titles are in sentence case, and that all species and gene names are properly italicized. Second, many of the main figures, which currently occupy a full page in landscape oreintation, will need to be redone to meet PLOS Genetics figure requirements, which can be found at  

In particular, once figures are redrawn to fit manuscript page width, you will find that fonts are so small as to be unreadable. Again, please see the above guidelines for font requirements.

Please modify the manuscript according to these review recommendations. Your revisions should address the specific points made by each reviewer.

Yours sincerely,

Michael Lichten, Ph.D.

Academic Editor

PLOS Genetics

Kirsten Bomblies

Section Editor

PLOS Genetics

Reviewer's Responses to Questions

**Comments to the Authors:**

Reviewer #1: The authors have substantially answered my queries/suggestions. One issue remains.

2. Hotspot conservation.

I appreciate the answer provided in the author response, however, I feel that this general concept needs clearer handling in the text (before lines 249 onwards). Currently this section rolls on directly from the preceding sentences without any introduction.

To help the reader, would it be possible to set the scene with a new paragraph here (the concept that hotspots may be conserved with evidence for why this is/was thought to be the case)? Then go on to present the tests/observations that the authors have performed (line 249 onwards).

Note: In general, I find the concept of why there would be crossover hotspot conservation within syntenic blocks between species perplexing. Certainly, in the absence of PRDM9, recombination initiates preferentially in promoter regions where there is low nucleosome occupancy. Yet, I don't understand why this generates any prediction for there to be a cross-species conservation in the *frequency* of recombination arising at orthologous/syntenic regions.

Even if the raw sequences were conserved (which I presume they are not), recombination frequency is far more complicated than that—being affected by broad-scale influences such as genomic location, chromosome number and size, proximity to other genomic elements, and by global recombination frequency.

Moreover, unless I am mistaken (and I hope the editor will clarify this with me if I am misrepresenting these prior findings) it has been shown by the editor of this manuscript that even an *identical* piece of DNA has very a different allelic recombination frequency depending on its genomic location (Goldman and Lichten 1996, Table 2).

As such, why then would *diverged* DNA sequences located in different genomic locations/contexts be expected to share similar frequencies? This seems to only make sense if one hypothesises that the regulators of recombination frequency are both insensitive to the inter-species DNA sequence changes and are also fortuitously constrained to act only within the limits of the syntenic blocks (and thus their influence is conserved). Both seem unlikely. Perhaps this can be explained, discussed and/or refuted?

(Perhaps a relevant detail that is missing from the manuscript is how large are the blocks of synteny that have been identified? It seems obvious that the larger the block, the more likely it would be to see similarities in frequency.)

Reviewer #2: The authors satisfactorily addressed all my minor points. However, they did not test the role of Mer3, the sole ZMM protein remaining, for sporulation nor spore viability. I understand that this experimental work could be saved for another manucript, but I still think it would be important to at least address the point of the possible role (or absence of role) of Mer3 in waltii (for instance in the Discussion).

**Have all data underlying the figures and results presented in the manuscript been provided?**

Reviewer #1: Yes

Reviewer #2: None

PLOS authors have the option to publish the peer review history of their article (what does this mean?). If published, this will include your full peer review and any attached files.

Reviewer #1: No

Reviewer #2: No

---

## [Editor Report · Decision Letter 2]

22 Dec 2022

Dear Dr Schacherer,

We are pleased to inform you that your manuscript entitled "Lessons from the meiotic recombination landscape of the ZMM deficient budding yeast Lachancea waltii" has been editorially accepted for publication in PLOS Genetics. Congratulations!

At the time you make those changes, please be sure to change "him" to "her" in the acknowledgements.

Yours sincerely,

Michael Lichten, Ph.D.

Academic Editor

PLOS Genetics

Kirsten Bomblies

Section Editor

PLOS Genetics

Comments from the reviewers (if applicable):

**Data Deposition**

http://datadryad.org/submit?journalID=pgenetics&manu=PGENETICS-D-22-00731R2

**Press Queries**

---

## [Editor Report · Acceptance letter]

2 Jan 2023

PGENETICS-D-22-00731R2 

Lessons from the meiotic recombination landscape of the ZMM deficient budding yeast Lachancea waltii 

Dear Dr Schacherer, 

We are pleased to inform you that your manuscript entitled "Lessons from the meiotic recombination landscape of the ZMM deficient budding yeast Lachancea waltii" has been formally accepted for publication in PLOS Genetics! Your manuscript is now with our production department and you will be notified of the publication date in due course.

With kind regards,

Livia Horvath

PLOS Genetics

On behalf of:
